# Deprivation of Sexual Reproduction during Garlic Domestication and Crop Evolution

**DOI:** 10.3390/ijms242316777

**Published:** 2023-11-26

**Authors:** Einat Shemesh-Mayer, Adi Faigenboim, Amir Sherman, Song Gao, Zheng Zeng, Touming Liu, Rina Kamenetsky-Goldstein

**Affiliations:** 1Institute of Plant Sciences, Agricultural Research Organization, The Volcani Institute, Rishon LeZion 7505101, Israel; shemeshe@volcani.agri.gov.il (E.S.-M.); adif@volcani.agri.gov.il (A.F.); asherman@volcani.agri.gov.il (A.S.); 2College of Horticulture and Landscape Architecture, Yangzhou University, Hanjiang District, Yangzhou 225012, China; gaosong@yzu.edu.cn (S.G.); 82101211030@caas.cn (Z.Z.); liutouming@caas.cn (T.L.)

**Keywords:** *Allium sativum*, flowering, genome, transcriptome, gene expression

## Abstract

Garlic, originating in the mountains of Central Asia, has undergone domestication and subsequent widespread introduction to diverse regions. Human selection for adaptation to various climates has resulted in the development of numerous garlic varieties, each characterized by specific morphological and physiological traits. However, this process has led to a loss of fertility and seed production in garlic crops. In this study, we conducted morpho-physiological and transcriptome analyses, along with whole-genome resequencing of 41 garlic accessions from different regions, in order to assess the variations in reproductive traits among garlic populations. Our findings indicate that the evolution of garlic crops was associated with mutations in genes related to vernalization and the circadian clock. The decline in sexual reproduction is not solely attributed to a few mutations in specific genes, but is correlated with extensive alterations in the genetic regulation of the annual cycle, stress adaptations, and environmental requirements. The regulation of flowering ability, stress response, and metabolism occurs at both the genetic and transcriptional levels. We conclude that the migration and evolution of garlic crops involve substantial and diverse changes across the entire genome landscape. The construction of a garlic pan-genome, encompassing genetic diversity from various garlic populations, will provide further insights for research into and the improvement of garlic crops.

## 1. Introduction

Garlic (*Allium sativum* L.) is an ancient vegetable and medicinal crop. Similar to onion, the wild ancestor of garlic has never been found in nature [1,2]. The current hypothesis of garlic’s domestication states that in Central Asia, wild garlic was collected and used by nomadic tribes, then Silk Road traders and Scythian travelers introduced these tasty plants to Southern Europe, the Mediterranean basin, India, and China; later, garlic was spread worldwide [3]. The large geographical distribution and the adaptation to various climatic conditions of garlic crops resulted in many varieties with specific morphological, physiological, and flavor traits. Similar to other crops, during garlic’s domestication and further crop evolution, only suitable plants were selected for the next generations, and much of the genetic diversity was probably lost [4].

Human selection for early maturation and large garlic bulbs deprived the developing reproductive organs of nutritional supplies, hence garlic domestication is associated with a complete loss of fertility and seed production [3]. Today, commercial garlic production is based exclusively on vegetative propagation, and garlic breeding is limited to selection from established genetic variation [5]. In vegetatively propagated crops, e.g., yams (*Dioscorea* spp.), bananas (*Musa*), or potatoes (*Solanum tuberosum*), the loss of seed dispersal and changes in plant architecture are significant markers of domestication [6,7]. Similarly, in garlic, phenotypic traits, the mode of reproduction, plant morphology, and environmental adaptations changed dramatically during domestication and crop evolution in various climatic regions [8].

The complete loss of flowering ability and seed reproduction during a relatively short time period of about 10,000 years [3,9] represents a fascinating example of fast crop evolution under human selection. It is still not clear, however, whether these phenotypic differences ensued from genomic mutations or differential gene expression. Currently, two main approaches facilitate identifying the genetic mechanisms of adaptation to domestication: the bottom-up approach uses population genetic and bioinformatics analyses to identify potentially adaptive genes, and the top-down approach applies genetic and transcriptome analysis to identify candidate genes associated with physiological traits [10].

In the framework of the breeding program in Israel, we collected and reproduced hundreds of garlic genotypes from various climatic zones varying in their reproductive traits. A large number of flowering genotypes producing viable reproductive organs and seeds were found in Central Asia. Further morpho-physiological studies showed that in some garlic genotypes, flower initiation and bolting (elongation of flower scape) are completely impaired, while others develop small topsets (bulblets) within the inflorescence which compete with flowers for nutrients. Moreover, even if individual flowers succeed in differentiating, seed production might be compromised by male or female sterility, tapetum degeneration, and pollen abortion [11,12,13].

Similar to many other perennial monocots, *Allium* species possess a large genome size (7–32 Gb) [14]. Despite its domestication, garlic has maintained its ploidy level (2n = 2x = 16), and the diploid garlic nuclear genome is estimated at 15.9 Gbp, 32 times larger than the genome of rice [15]. Therefore, full genome sequence is challenging, and previous garlic genetic studies were based on transcriptome and candidate gene analyses. The generation of transcriptome catalogs [16,17] resulted in the development of simple sequence repeats (SSRs) that can be used for genetic studies, mapping, and fingerprinting [18]. Organ-specific transcriptome analysis displayed significant variation in gene expression between different plant organs, while the highest number of specific reads was found in the inflorescences and flowers [17].

The recent publication of the full genome sequence [16] of garlic enabled in-depth research on garlic biology [13,19]. Here, we provide phenotypic, genomic, and transcriptomic analyses of several genetic groups of garlic. About 200 accessions with different flowering traits were reproduced in common garden experiments in Israel during several seasons. We selected 41 accessions with clear phenotypic variances in bolting and flowering under the same environmental conditions and performed their genomic and transcriptome analyses. Using whole-genome resequencing (WGRS), we estimated genomic differences between garlic populations. We postulate that the loss of flowering ability during garlic crop evolution is associated not only with specific alterations in the flowering genes, but also with large and multiple mutations in the entire genome landscape. Our results suggest that the deprivation of garlic flowering was not linear but proceeded in several directions, and numerous genetic changes in flowering pathways occurred in a relatively short time period during crop evolution.

## 2. Results and Discussion

### 2.1. Nucleotide Variation in Garlic Genotypes

The sequencing of 41 genotypes resulted in 305.4 million SNPs and 21.9 million indels. Quality filtering resulted in 63.6 million SNPs and 4.5 million indels. Based on nucleotide substitutions (SnpEff), the SNPs were classified as transitions (Ts) (purine–purine and pyrimidine–pyrimidine) or transversions (Tv) (purine–pyrimidine and pyrimidine–purine). The Ts/Tv ratio in the SNPs variants was 1.85. The number of SNPs in eight chromosomes (Chr1–Chr8) ranged from 6 M to 9.9 M SNPs, with an average of 7.9 M SNPs per chromosome. The number of indels in eight chromosomes (Chr1–Chr8) ranged from 433,128 to 709,131, with an average of 567,755 indels per chromosome. The average variant rate was one SNP every 223 bases and one indel every 3128 bases. Most of these SNPs (91.66%) were located in intergenic regions, 3.05% were in genic regions, and the remaining 5.28% were in upstream and downstream regions (Appendix A). Considering those in the genic regions, 321,662 SNPs were in exonic regions, 1,711,050 SNPs were in intronic regions, and 19,778 SNPs were in splicing regions. In total, 208,492 SNPs generated amino acid mutations, including non-synonymous substitution, stop gain, and stop loss (Appendix A).

### 2.2. Population Structure and Diversity

Phenological and morphological observations in the common garden experiments divided garlic accessions into four main groups. However, further analysis of the population genome structure and phylogenetic relationships between 41 accessions using the ADMIXTURE program and NJ phylogenetic algorithm indicated six distinct groups (Figure 1a). Seven genotypes out of forty-one did not align into these clusters, and rather included genomic elements of two, three, and even four different groups (Figure 1b).

In general, phenotypic division coincided with the phylogenetic tree, but, in addition, the genomic analysis divided bolting flowering genotypes into three clusters—A, B, and C. Moreover, cluster A consists of two sub-clusters, both producing fertile flowers and seeds.

The flowering genotypes from clusters A, B, and C were introduced either directly from Central Asia or via the gene banks in Europe (Appendix A). They consist of bolters with long flower stems and flowers in the inflorescences (Figure 2a,b). Clusters A and B include the most fertile accessions, producing seeds. Some genotypes are male sterile, while others produce pollen with different levels of fertility [20]. Genotypes of cluster C develop flowers, but most of them degenerate just before anthesis. Biomorphological analysis of the clusters A–C corresponds to a primitive and diverse Longicuspis group originating in the continental climate of Central Asia, with cold winters and long hot summers [8,21]. Central Asia is considered the primary center of garlic domestication, but AFLP analysis confirmed that the Longicuspis group is not homogeneous [22]. This gene pool is less well known outside of Central Asia, and may contain genes of interest for future use in genetic studies and crop improvement programs.

Cluster D includes semi-bolting genotypes. Their apical meristem terminates the vegetative stage but does not produce reproductive meristems or individual flowers primordia. Instead, only one or two large topsets are formed on the short flower stem [23] (Figure 2c,d). Although introduced to our collection from different locations (Appendix A), these genotypes probably have a common origin in the Mediterranean area. For instance, the variety ‘Shani’ is grown in Israel, while ‘Palestinian’ was introduced from the USA, but originated in the Mediterranean (A. Drucker, Pers. Comm, 2017). The growth conditions in the Mediterranean areas, with warm winters and short days, are in contrast to the cold winters and long summer days in Central Asia. Therefore, in addition to crop evolution under human cultivation, these genotypes adapted their physiological traits to the new environmental conditions. This adaptation might result from the long selection of the original types introduced from Central Asia via the Silk Road about 5000 years ago 5000? Five thousands [24]. Another assumption is that this type of garlic was directly domesticated and evolved in the Mediterranean region, recognized as the second center of garlic diversity [9,25,26].

The genotypes of Cluster E (Figure 1) produce long flower stalks and flower meristems, but flower buds degenerate prior to full differentiation, and only a few reach anthesis. The mature inflorescence contains small or large topsets (Figure 2e,f). This type has the largest geographical distribution in Eurasia, and the full genome sequence of the Chinese variety from this cluster was recently assembled [16].

Finally, non-bolting genotypes of Cluster F (Figure 1) were introduced from Europe (Appendix A) and did not produce flower stalks under our local conditions. Microscopic records show that at the end of the season, the meristem terminates leaf development and forms small storage scales instead of inflorescence [27] (Figure 2g,h).

### 2.3. Genetic Diversity and Plant Architecture in Genomic Landscape

Clusters A, B and C belong to the same morpho-biological type and represent the flowering genotypes of garlic. Therefore, we selected Cluster A as a basic group that also represents both Clusters B and C. Further comparisons were made between Cluster A and Clusters D, E, and F, which vary significantly in their morphological, physiological and genetic traits.

The data from the WGRS of these garlic subpopulations were validated against the garlic genome [16]. The allele frequencies were calculated in each SNP position individually in the four clusters, using an in-house Perl script. Positions showing an alternative allele (vs. the reference genome) frequency higher than 0.95 were considered as SNPs.

We found that only 521 SNPs (0.005%) were common in all clusters, while 3,438,212 SNPs were unique to Cluster D, 1,290,571 were unique to Cluster F, and 1,136,253 were unique to Cluster A. Surprisingly, only 2205 of unique SNPs, were found in Cluster E. This probably reflects the fact that the accession used for the sequencing of the reference genome (garlic cultivar ‘Ershuizao’ from China) [16] belongs to Cluster E, and therefore, it is not significantly different. Further genome sequencing of the heterogeneous biogeographical types and groups of garlic will provide an opportunity to exploit hidden variations in flowering and other useful traits [28].

The nucleotide diversity (π) and fixation index (*F*_ST_) were calculated to evaluate the genetic diversity between the four clusters and their genetic differentiation. Analysis of genetic diversity and differentiation using *F*
_ST_ indicated similar levels of differentiation between Clusters A and F (*F*_ST_ = 0.3447), E (*F*_ST_ = 0.349), and D (*F*_ST_ = 0.345). The highest genetic differentiation was found between Clusters D and E (0.435) and F (0.422). The nucleotide diversity was by far higher in Cluster A than in all other groups (Figure 3).

In order to estimate the genetic differences associated with reproductive traits, population diversity was assessed based on 500 kb sliding windows. Estimates of *F*_ST_ can identify regions of the genome that have been the target of natural or human selection, and comparisons of *F*_ST_ from different parts of the genome provide insights into the demographic history of populations [29].

To identify potential selection signatures and candidate genes involved in florogenesis, the accessions of three clusters D, E, and F were scanned using *F*_ST_ vs. cluster A and GO enrichment analysis. The regions with an *F*_ST_  ≥  0.5 (top 1.8–2.6%) were considered potential regions under selection (Appendix A). In total, 2555 candidate genes were identified in at least one of those windows with high *F*_ST_ values, 794 in D vs. A, 1148 in E vs. A, and 1192 in F vs. A. GO enrichment analysis showed that these genes were enriched in multiple terms associated with biosynthesis, development, metabolic processes, defense and stress responses (Figure 4, Figure 5 and Figure 6).

Recently, Jia et al. [30] reported that at least 135 SNPs were associated with bolting and flowering in garlic. We detected significant differences between clusters with different reproductive abilities and plant architecture, not only in the processes directly connected to plant growth and florogenesis, but also in hormonal signals, defense, and response to stress. The comparison between Clusters A and D reveals enrichment in 496 biological processes that are associated with garlic crop evolution (Figure 4). The most significant differences were found in the plant development (e.g., flower, root, leaf development, and meristem organization), stress response, and metabolism. Clusters A and D differ in both bolting ability and inflorescence development. Flower meristems are not initiated in Cluster D, and only one or two large topsets are produced following to meristem transition. We imply that the selection signatures are associated with this core change in the developmental cycle and completely deprived sexual reproduction in Cluster D.

The estimation of population and GO enrichment analysis between clusters A and E reveals 623 biological processes possibly involved in the selection process in various climatic zones. The accessions of both clusters produce long flower stalks and initiate flower meristems, and therefore the main morphological differences between these clusters relate to flower abortion during inflorescence differentiation (Figure 2). It is largely accepted that during garlic crop evolution, human selection prevented sexual reproduction for the benefit of larger bulbs [3]. Our GO analysis of SNP-containing genes and their differences between Clusters A and E implies that the process of flower deterioration and topset formation might be associated with numerous molecular events. Most of them belong to the development, biosynthesis/metabolism, and defense mechanisms and are identified as reproductive processes (e.g., meristem, gynoecium, pollen, seed coat, root and flower development) or processes associated with cell division, photosynthesis and light perception (Figure 5). On the other hand, we found significant enrichment in metabolic processes and the biosynthesis of jasmonic acid, proteins, lignin, sulfur, carbohydrates, and more. DNA methylation, replication, and phosphorylation processes also differ on the genomic level between the two clusters (Figure 5).

The enrichment of GO terms between Cluster A and non-bolting Cluster F reveals 601 biological processes, with an abundance of the patterns involved in stress response and defense mechanisms. The development and metabolic processes are also enriched (Figure 6). The accessions in cluster F do not bolt, but the meristem terminates, and therefore the molecular events might be associated with bolting and florogenesis arrest in the Cluster F.

We imply that during garlic introduction and crop development in climates different from that in Central Asia, the genotypes adapted to various environmental conditions: short days and warm winters in the Mediterranean, cooler summers in Europe and the subtropical climate in India. Therefore, during the evolution of garlic crop, genetic modification occurred not only in the genes directly regulating bolting or/and florogenesis, but also in the larger genomic landscape, including stress adaptation and defense mechanisms.

### 2.4. Genomic and Transcriptomic Analysis of Flowering-Related Gene Families

A total 179 inter-genome variations, representing 82 genes, were scored for the high and moderate impact on plant development, and several specific gene families related to florogenesis were analyzed at both the genomic and transcriptome levels (Figure 7 and Figure 8).

Meristem transition and florogenesis are coded in plants by a multifaceted gene network of vernalization, photoperiod, gibberellins, and autonomous signaling modules [31]. However, considerable differences exist in vernalization perception between plant families, e.g., Brassicaceae [32], cereals [33], Rosaceae [34], and other lineages.

In garlic, the temperature is the main environmental factor regulating the flowering process. Low temperatures stimulate cascades of developmental mechanisms in several genetic pathways. The transcripts annotated as vernalization genes play a central role in the flowering pathway’s interplay [13,35]. Our genome and transcriptome analyses indicate that several variants of *VERNALIZATION INSENSITIVE 3* (*VIN3*), a key gene in temperature perception and plant response to cold [36], are located in garlic chromosomes 3, 4 and 6. At least 15 variants of *VIN3* with a high or moderate SNP effect differ in allele frequency between garlic clusters (Figure 7a). At the same time, transcriptome analysis reveals only three differentially expressed *VIN3* transcripts (Figure 7b). Since phenotypic expression and florogenesis of garlic are strongly affected by pre-planting cold treatment and/or winter temperatures [37], we argue that the genome mutations in *VIN3* variants might provide one of the turning points in divergent evolution in garlic subpopulations.

In model plants, *VIN3* regulates the floral repressor *FLOWERING LOCUS C* (*FLC*). FLC-related proteins repress floral integrators, including *FT, FD*, and *SOC1,* regulate the timing of flowering, and play an essential role in meristem transition [32,38].

However, since flowering evolved in various climates several times in plant history, regulatory mechanisms vary. Thus, in wheat, maize, and rice, vernalization is regulated through *VRN1, VRN2,* and *VRN3* (or *FLOWERING LOCUS T*) [39]. Monocots may be *FLC*-independent for flower initiation [40]. Our genomic analysis did not discover *FLC* homologs in garlic, but homologs of flower repressors *FLC EXPRESSOR (FLX), UPSTREAM OF FLC (UFC)*, and *FRIGIDA (FRI)* were found, and their variants in several chromosomes show high variability between garlic subpopulations. At least six *UFC* homologs, located in chromosomes 2, 6, and 7, show structural differences between flowering and non-flowering genotypes (Figure 7c). The homologs of these genes were also found in *Gladiolus* [41]. In *Arabidopsis*, *UFC* is adjacent to the endogenous *FLC* and is repressed by vernalization [42]. Therefore, local mutations in the floral repressors can certainly delineate differences in bolting and flowering genotypes. The differences are even more evident at the transcriptome level (Figure 7d). Interestingly, the transcripts of some genes with higher mutations intensity (e.g., *FRI*_Asa7G05905; *UFC*_Asa2G03096; *UFC*_Asa6G02411) do not display differential expression. Some floral repressors were upregulated in all subpopulations (e.g., *FLX*_Asa2G07102.1; *FLX*_Asa6G07053.1; *MAF1*_Asa3G01998.1; *FRI*_Asa7G05899.1). The second group of vernalization-related genes were downregulated in all genotypes (e.g., *FRI*_Asa2G05168.1; *FRI*_Asa8G06068.1; *UFC*_Asa3G01273.1). Finally, some genes (e.g., *FLX*_Asa0G02781.1; *UFC*_Asa2G04587.1) were upregulated mostly in non-flowering semi-bolters of Cluster D. We conclude that putative flower repressors with different structures might act in garlic as *FLC* orthologs, but we are still not able to identify these genes.

We discovered a large abundance of multiple gene copies and numerous SNPs in the genes of the circadian clock. It was already shown that during vernalization phytochrome B (PHYB), a major light sensor of the circadian rhythm, acts as a thermosensor [35,43], while EARLY FLOWERING 3 (ELF3) transmits thermal information [44]. In general, the PHYB family mediates not only developmental responses from seed germination to flowering, but also abiotic stress acclimation responses and plant resistance to environmental stress [45]. We found high mutation intensity in one *PHYB* variant (*PHYB*_Asa2G00847) in all comparisons, while the other seven *PHYB* variants showed high mutation intensity in only one or two comparisons (Figure 8a). Other genes associated with circadian rhythm, e.g., *PHYTOCHROME INTERACTING FACTOR 3 (PIF3), ELF3, LATE ELONGATED HYPOCOTYL (LHY)*, also show high polymorphism on the genome level. The modifications and mutations in the genes of circadian clock might affect phenotypic differences between garlic subpopulations and cardinal changes in bolting and flowering ability. At the transcriptome level, differential expression of the circadian clock genes might provide the additional mechanisms for flowering regulation (Figure 8b). Notably, at least eight *Suppressor of PHYB (SPAB*) genes are expressed differentially between four garlic clusters (Figure 8b), but SNP analysis did not reveal variation in these genes at the genome level (Figure 8a).

Multiple variants of the genes associated with the photoperiod pathway, *GIGANTEA (GI), CYCLING DOF FACTOR 1 (CDF1)* and *CONSTANS (CO)* are located in several chromosomes and show structural differences between garlic subpopulations (Figure 8c). We found large differences between four garlic subpopulations in the expression of photoperiod genes (Figure 8d). Cluster analysis of the differential expression suggests that the genotypes of the cluster D that are adapted to a short photoperiod and grow in the Mediterranean climate during its warm winters diverge from other clusters.

Surprisingly, we did not find significant inter-population variation in the structure and expression of the meristem-identity genes. The *FLOWERING LOCUS T (FT)/TERMINAL FLOWER 1 (TFL1)* family of phosphatidylethanolamine-binding protein (PEBP) genes is fundamental to plant development, the regulation of flowering time and plant architecture, including monocot geophytes [46,47], and is highly conserved across the plant kingdom [48]. The role of PEBP and *LEAFY (LFY)* homologs in meristem transition in garlic was reported [13]. However, within 26 PRBP genes present in the garlic genome, only one *FT* homolog in chromosome 4 exhibits SNP polymorphism between garlic subpopulations, and four *FT* homologs in chromosomes 6, 7, and 8 were differentially expressed (Figure 8e,f).

It is possible that during crop evolution, cardinal changes occurred in the mechanisms of flower induction, especially vernalization and the circadian clock, while meristem identity genes are conserved and function similarly in different garlic populations.

### 2.5. Crop Migration and Evolution Led to the Major Changes in Garlic Biology

Based on the biogeographical analysis, one hundred years ago Vavilov [21] hypothesized that garlic originated from Afghanistan, Tajikistan, Uzbekistan, and western Tien-Shan about 10,000 years ago, while the Mediterranean area is a secondary center of garlic diversity. Later studies of interspecific differentiation based on isozyme and RAPD markers categorized garlic into four major groups and two subgroups [8]. Here, we employed genomic and transcriptomic tools to depict possible means of garlic domestication, crop evolution, and flowering depreciation (Figure 9).

The most heterogeneous and primitive garlic types with bolting and flowering ability were collected in Central Asia, one of the original habitats of garlic’s ancestor(s) [49,50]. Although in our living collection, these genotypes do not differ phenotypically, phylogenetic analysis allocated them to three clusters A, B, and C (Figure 1). These clusters mirror the garlic domestication process: bolting genotypes vary in the quality of flowers, flower sterility, and seed production. We assume that during domestication, the plants were initially cultivated in Central Asia and selected for larger bulbs. Later, traders conveyed garlic bulbs to different climatic regions [3], where crop evolution resulted in complete flowering depreciation and changes in the annual cycle and plant architecture (Figure 9).

Our genomic analysis supports an assumption that the Mediterranean group was derived from the West Asian group very early and possesses unique genetic traits [51]. The Mediterranean cluster D notably diverges from the other groups in its genome structure, phenological cycle, plant architecture, and flowering ability (Figure 1 and Figure 2). Members of this cluster do not initiate reproductive organs under any environmental conditions [23].

The migration of garlic crops from the main center to the relatively cool regions in Europe and Asia resulted in changes in the inflorescence structure and the development of numerous topsets in bolting varieties. Cluster E is the most abundant type in garlic cultivation, found in all continents. Flowers are initiated in these bolting genotypes but are aborted when topsets develop in the inflorescence.

Cluster F derived from the same phylogenic lineage, but evolved in the warm regions of India (Figure 1a and Figure 9). In addition to the genetic changes, bolting in this cluster is deprived by environmental regulation.

We conclude that the loss of flowering ability during garlic domestication and further crop evolution in different climates is associated with large and multiple mutations in the entire genome landscape. This does not comprise one or a few mutations in specific genes, but comprehensive changes in the genetic regulation of the annual cycle, architecture and environmental requirements. These mutations in the specific genes associated with vernalization and circadian clock had a major impact on crop evolution. Since the garlic genome is extremely large and highly repetitive [16], the homologs of key genes involved in florogenesis and stress adaptation are located in all chromosomes, and variants are expressed differentially under changing environments. Therefore, the flowering ability, stress response, and metabolism are regulated at both the genetic and transcription levels.

The recent assembly of the complete genome of garlic provided a huge step forward in garlic research [14,52]. However, the current reference genome reflects only one biogeographical group of garlic that belongs to cluster E in this study. We argue that, similar to other crops [19], the construction of a pan-genome of garlic that embraces variable genetic diversity will provide comprehensive information for further garlic research and improvement.

## 3. Materials and Methods

### 3.1. Plant Material and Phenology

The garlic collection in the Agricultural Research Organization, belonging to the Volcani Center in Israel, contains more than 200 genotypes introduced from different regions or reproduced from true garlic seeds during the last 20 years. For this study, we selected 41 genotypes with marked phenotypic differences in their floral development (Figure 10). For three seasons (2020–2022), plants were grown in common garden experiments in the R&D Station in Avnei Eitan, Northern Israel. Common agriculture practices were applied during the growing season. The morphology of five inflorescences in each accession was recorded during growing season under a stereoscope (Zeiss Stemi 2000-C, Zeiss, Germany).

According to their morphological and phenological features, we divided the genotypes into four groups: (1) bolting and flowering, with an inflorescence that contains developed flowers; (2) bolting, with an inflorescence that contains small topsets, but all or most of the flowers aborted; (3) semi-bolting, producing short flower stems and a few large topsets, but no flower meristems and flowers; and (4) non-bolting (Figure 10; Appendix A).

### 3.2. Genome Sequencing

During the vegetative growth, the youngest leaves were collected from five plants of each of the 41 genotypes and dried at 60 °C for 24 h. Then, 20–1000 mg leaf samples from each genotype were shipped for the further analysis in China. Genome sequencing was performed at Shanghai OE Biotech. Co., Ltd. (Shanghai, China). The DNAsecure Plant Kit (TIANGEN) was used for the extraction of the genomic DNA (gDNA) from the dry leaves sent from Israel. After determining the quality and quantifying the concentration of gDNA, the data were used to construct a sequencing library via the TruSeq Nano Sample Prep Kit (Illumina Inc., San Diego, CA, USA). In brief, about 1.5 µg of gDNA was fragmented by sonication to a size of approximately 350 bp, and then, the generated DNA fragments were end-polished, A-tailed, and ligated with the full-length adapters. After PCR amplification, the corresponding PCR products were purified using the AMPure XP bead system. The size distribution of generated libraries was analyzed using the Agilent2100 Bioanalyzer (Agilent Technologies, Santa Clara, CA, USA), and each library was sequenced using the Illumina HiSeq X platform, generating reads with a 150 bp length.

### 3.3. Variation Calling

For SNP and indel analysis, the filtration of low-quality paired reads (FastQC package v.0.11.9; https://www.bioinformatics.babraham.ac.uk/projects/fastqc (accessed on 20 November 2023)) was performed, and the high-quality reads were aligned with the garlic reference genome (GCA_014155895.2) via the Burrows–Wheeler Aligner (BWA) software v.0.7.8 [53], following the default parameters. The results of alignment were converted into a BAM format and sorted via SAMtools v.1.3 [54]. The genomic variants from each accession were identified using the Bayesian approach implemented in the SAMtools package and filtered using the following criteria: genotype quality of each individual less than 5, depth of each individual less than 3, and QUAL less than 20.

### 3.4. Functional Annotation

After filtering minor allele frequency (MAF), the SNPs (MAF < 0.05 and minimum 3DP per individual) were retained for subsequent analysis. The SnpEff package [55] was used for SNP annotation based on the gene model reference, and to categorize the SNPs according to their positions, including intergenic, intronic, exonic, splicing sites, downstream and upstream regions.

A phylogenetic tree was constructed using TASSEL software v. 4.0 [56] based on the neighbor-joining (NJ) method and visualized using FigTree. The population structure of the accessions was analyzed using Admixture software v.1.23 [57]. Nucleotide diversity (π) was used to estimate the degree of variability within each subpopulation, and the fixation index (*F*_ST_) among subpopulations was used to explain genetic differentiation. The π and *F*_ST_ were calculated using VCFtools v.0.1.14 [58] based on 500-kb sliding windows, using genome-wide SNPs. The allele frequencies were calculated in each SNP position individually in each group using an in-house Perl script.

Genomic windows with high *F*_ST_ (>0.5) values were considered as potential selection regions. The genes in the potential selection regions were analyzed for Gene Ontology (GO) enrichment using the KOBAS web tool [59]. The main molecular and biochemical pathways associated with interspecific variation were assessed using the REVIGO software v. 1.8.1. The multidimensional scaling (MDS) technique was utilized to reduce the dimensionality of semantic similarities among GO terms related to biological processes. The axes in the plot have no intrinsic meaning. Semantically similar GO terms remain close together in the plot. Bubble size represents the *p*-values of the GO term; bubble color represents the number of annotations for each GO term [60].

### 3.5. Transcriptome Analysis

For transcriptome analysis, roots, foliage leaves, the basal plate, the young floral scape (if present), and cloves were collected in three replicates from eight genotypes, representing all phenotypic groups (Appendix A), according to the variable development stages of each genotype. Tissue samples were collected in March-May 2016, immediately dipped in liquid nitrogen and stored at −80 °C. Total RNA was extracted using the CTAB protocol [61], and extract quality was assessed using an Agilent 2100 Bioanalyzer (Agilent, Santa Clara, CA, USA). Only extracts with a minimum RNA integrated value of six were used for analysis. For each genotype, 1000 ng of RNA of each replicate of each organ was mixed in a common tube and used for further RNA sequencing. Library preparation and sequencing were performed at the Genome Center, Life Sciences and Engineering, Technion, Israel Institute of Technology, Haifa, Israel.

Clean reads from the transcriptomes [16,17,23] were mapped to the reference genome of *Allium sativum* (GCA_014155895.2) using STAR software v. 2.7.1a [62]. We also analyzed published transcriptome catalogs of the flowering genotype #87 and the semi-bolting ‘Shani’ cultivar [19,45]. Gene abundance estimation was performed using Cufflinks v.2.2 [63], combined with gene annotations [16]. Integrated gene-expression values were computed as fragments per kilobase of the transcript per million mapped reads (FPKM). Transcript FPKM under 10 were reduced, and average FPKM values were used to create a heat map.

### 3.6. Data Mining for Flowering Genes

Based on the literature and our previous research [13,17,35,64,65,66], we surveyed flowering pathways and specific genes conceivably involved in the various stages of garlic floral development, e.g., meristem transition, floral scape elongation, differentiation of flowers, and floral organs. This survey resulted in a list of 543 candidate genes (Appendix A). This list was further analyzed using genomic and transcriptomic tools.

SnpEff algorithm [48] was used to find 46,588 SNP associated with 543 gene accessions, and the SNPs were filtered as follows: (1) the allele frequencies were calculated in each SNP position individually in each group using an in-house Perl script; (2) selection for SNP differences in the allele frequency Δ > 0.7 in at least one comparison between the groups resulted in 4750 SNPs; and (3) among all of the variants detected, 16 (0.33%) and 163 (3.43%) were predicted to produce high- or moderate-impact phenotype changes, respectively. Taken together, the applied filtrations resulted in 179 SNPs categorized into functional groups according to their annotation and were used as a basis for analysis and comparison of the allele frequencies between the phenotypic groups.

## Figures and Tables

**Figure 1 ijms-24-16777-f001:**
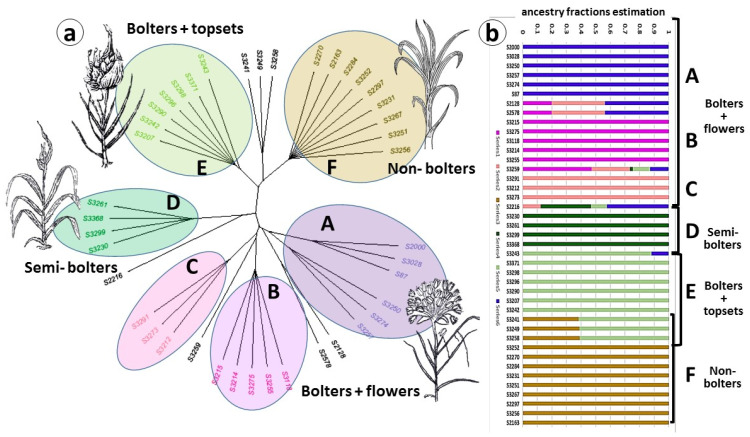
Population structure and phylogenetic relationships among 41 garlic accessions. (**a**) NJ phylogenetic tree of garlic genotypes, clustered into six main groups, based on genome variants. Seven genotypes did not fit with any of the main clusters. Clusters A, B and C represent one morpho-biological type; (**b**) population structure using ADMIXTURE software v.1.23. Note genotypes with genetic elements of several phenotypic groups/genetic clusters.

**Figure 2 ijms-24-16777-f002:**
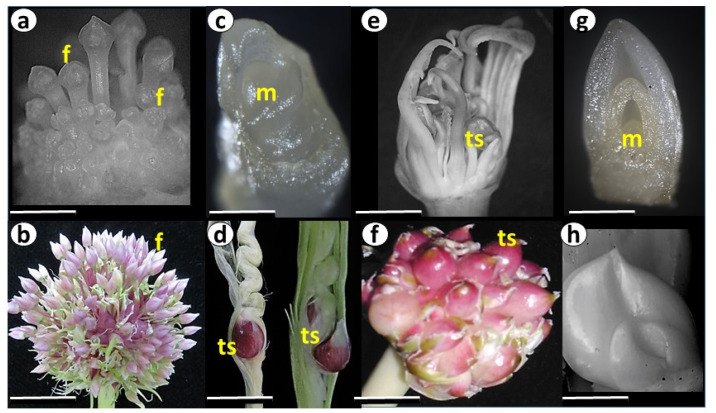
Meristem and inflorescence development in garlic genotypes. Plant morphology was recorded during three growing seasons under common garden experimental conditions. f—flowers; ts—topsets; m—apical meristem. (**a**,**b**) Primordial inflorescence with flower buds and inflorescence at anthesis, cluster A, genotype #87; (**c**,**d**) the apical meristem does not initiate flowers and only a few topsets develop, cluster D, genotype #3261; (**e**,**f**) primordial inflorescence with young flower buds and topsets and mature inflorescence with topsets, cluster E, genotype #3242; (**g**,**h**) apical meristem does not initiate reproductive organs and growth terminates with bulbing, cluster F, genotype #2163. Scale bars: (**a**)—0.75 cm; (**b**)—1.5 cm; (**c**)—150 µm; (**d**)—3 cm; (**e**)—1 cm; (**f**)—1.5 cm; (**g**)—200 µm; (**h**)—3 cm.

**Figure 3 ijms-24-16777-f003:**
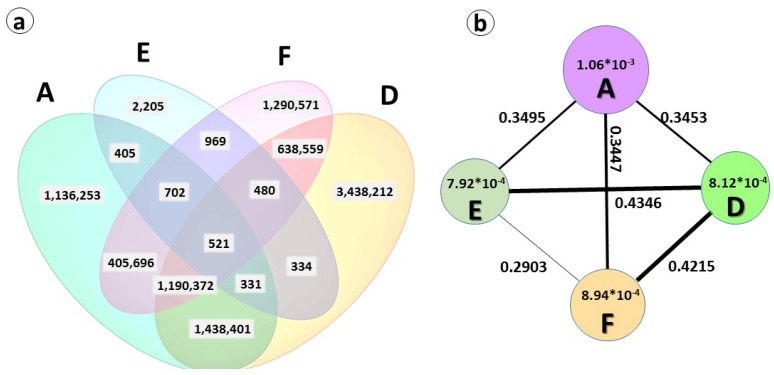
Genetic diversity and differentiation analysis of four garlic subpopulations, Clusters A, D, E, and F (see Figure 1). (**a**) Venn diagram summarizing the quantity of unique and common SNPs between each group and the reference genome (>0.95 in the alternative allele frequency); (**b**) nucleotide diversity (π) and fixation index (*F*_ST_) across subpopulations. The value in each circle represents the nucleotide diversity for each subpopulation/cluster, and the circle size indicates the π magnitude; the value on each line indicates the genetic divergence between two subpopulations, and the line width indicates the *F*_ST_ magnitude.

**Figure 4 ijms-24-16777-f004:**
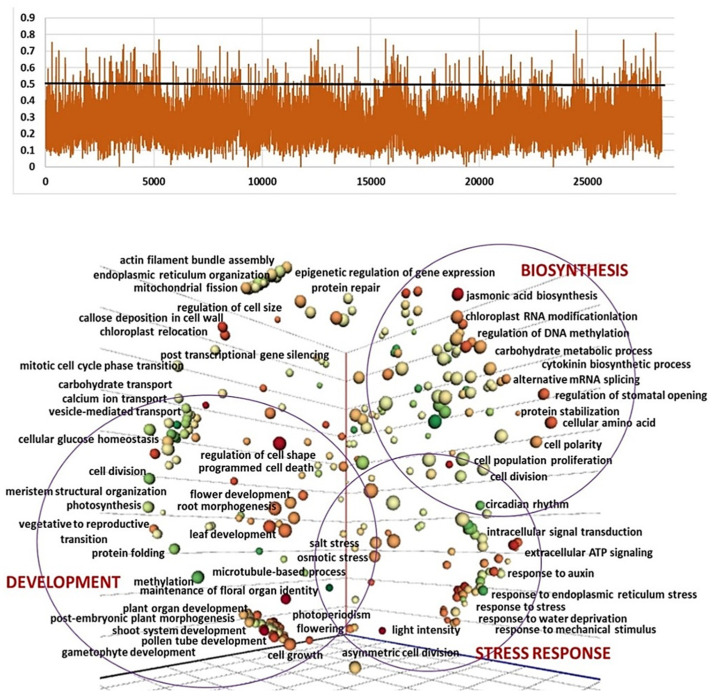
Selection signatures in the garlic genome in the comparison between Cluster A (bolting and flowering genotypes) and Cluster D (semi-bolters with large topsets at the top of the flower stem). (**a**) Based on the *F*_ST_, 512 regions belonging to the top 1.8% above the cut-off at 0.5 are defined as potential regions under selection. Peaks represent higher variation between clusters. (**b**) GO analysis of genes in genomic windows with significantly high *F*_ST_ (>0.5) of clusters A vs. D. The multidimensional scaling (MDS) technique is utilized to reduce the dimensionality of semantic similarities among GO terms related to biological processes. The axes in the plot have no intrinsic meaning. Semantically similar GO terms remain close together in the plot. Bubble size represents the *p*-values of the GO term; bubble color represents the number of annotations for GO term.

**Figure 5 ijms-24-16777-f005:**
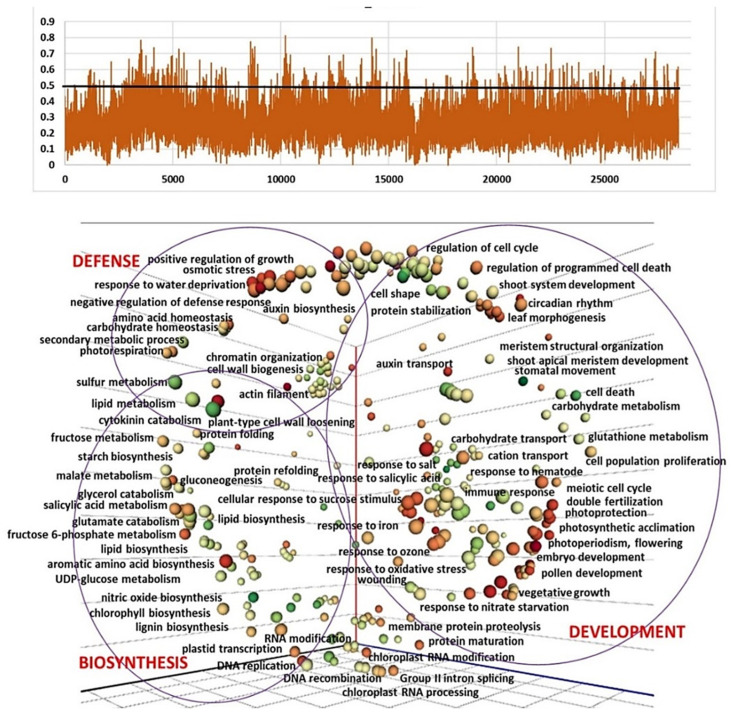
Selection signatures in the garlic genome in the comparison between Cluster A (bolting and flowering genotypes) and Cluster E (bolters with topsets, aborted flowers in the inflorescence). (**a**) Based on the *F*_ST_, 694 regions belonging to the top 2.4% above the cut-off at 0.5 are defined as potential regions under selection. Peaks represent higher variation between clusters. (**b**) GO analysis on genes in genomic windows with a significantly high *F*_ST_ (>0.5) of clusters A vs. E. GO terms of the biological processes are presented. The multidimensional scaling (MDS) technique is utilized to reduce the dimensionality of semantic similarities among GO terms related to biological processes. The axes in the plot have no intrinsic meaning. Semantically similar GO terms remain close together in the plot. Bubble size represents the *p*-values of the GO term; bubble color represents the number of annotations for GO term.

**Figure 6 ijms-24-16777-f006:**
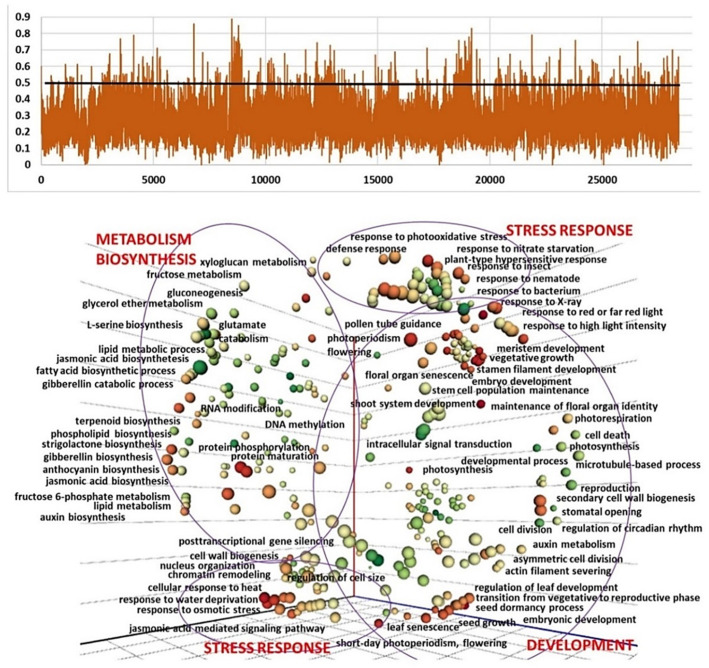
Selection signatures in the garlic genome in the comparison between Cluster A (bolting and flowering genotypes) and Cluster F (non-bolters). (**a**) Based on the *F*_ST_, 694 regions belonging to the top 2.4% above the cut-off at 0.5 are defined as potential regions under selection. Peaks represent higher variation between clusters. (**b**) GO analysis on genes in genomic windows with a significantly high *F*_ST_ (> 0.5) of clusters A vs. F. GO terms of the biological processes are presented. The multidimensional scaling (MDS) technique is utilized to reduce the dimensionality of semantic similarities among GO terms related to biological processes. The axes in the plot have no intrinsic meaning. Semantically similar GO terms remain close together in the plot. Bubble size represents the *p*-values of the GO term; bubble color represents the number of annotations for GO term.

**Figure 7 ijms-24-16777-f007:**
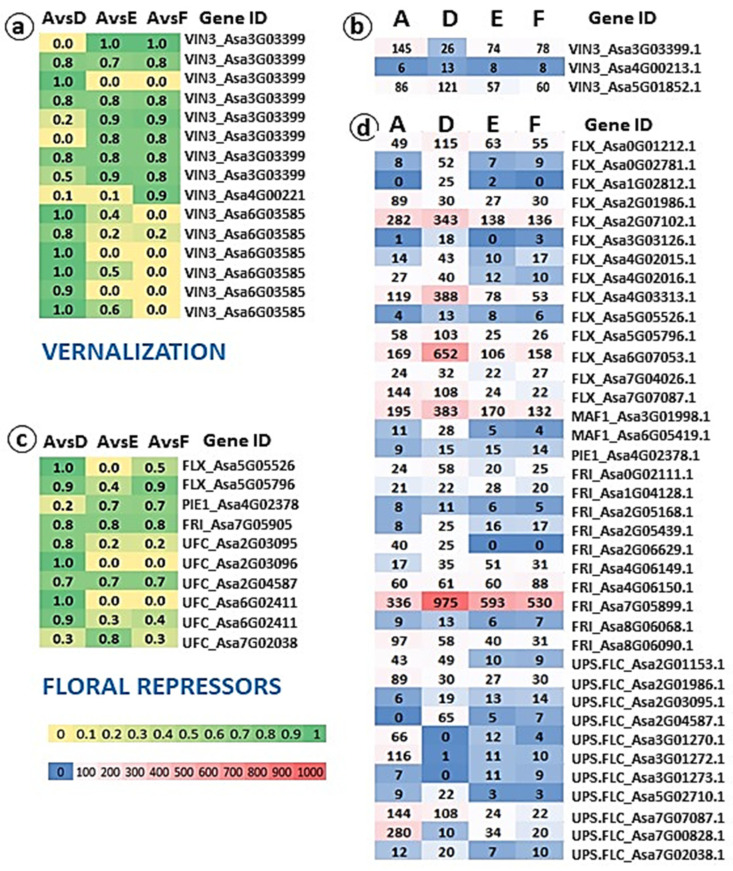
Genome and transcriptome analyses of the genes associated with vernalization and flower repression in four garlic subpopulations with different reproductive traits. Comparisons were made between Cluster A (bolting and flowering genotypes) and Clusters D (semi-bolters with large topsets at the top of the flower stem), E (bolters with topsets, aborted flowers in the inflorescence), and F (non-bolters). (**a**,**c**) Allele frequency map. Numbers represent the delta (Δ) between allele frequencies of each variant in each group comparison. Only variants with ‘high and moderate impact’ and differences in the allele frequency Δ > 0.7 in at least one comparison are included. The color scale ranges from yellow (low value) to green (high value). (**b**,**d**) Heat map of the expression patterns between four clusters. The expression values were computed as FPKM based on transcriptome analyses in each phenotypic group. FPKM values were used to create a heat map. The color scale on the heat map ranges from blue (low value) to red (high value).

**Figure 8 ijms-24-16777-f008:**
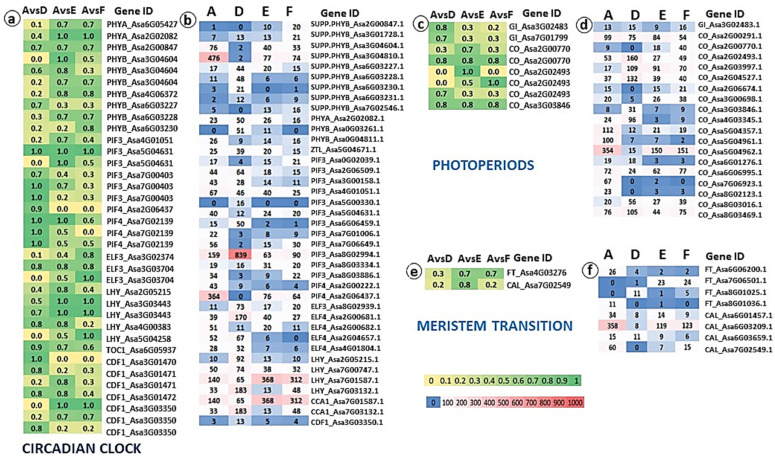
Genome and transcriptome analyses of the genes associated with the circadian clock, photoperiod, and meristem transition in four garlic subpopulations with different reproductive traits. Comparisons were made between Cluster A (bolting and flowering genotypes) and Clusters D (semi-bolters with large topsets at the top of the flower stem), E (bolters with topsets, aborted flowers in the inflorescence), and F (non-bolters). (**a**,**c**,**e**) Allele frequency map. Numbers represent the delta (Δ) between allele frequencies of each variant in each group comparison. Only variants with ‘high and moderate impact’ and differences in the allele frequency Δ > 0.7 in at least one comparison are included. The color scale ranges from yellow (low value) to green (high value). (**b**,**d**,**f**) Heat map of the expression patterns between four clusters. The expression values were computed as FPKM based on transcriptome analyses in each phenotypic group. FPKM values were used to create a heat map. The color scale on the heat map ranges from blue (low value) to red (high value).

**Figure 9 ijms-24-16777-f009:**
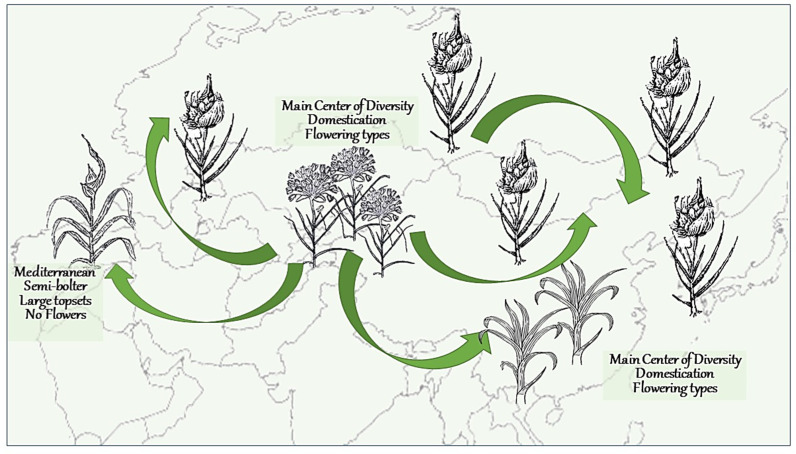
The proposed pathways of garlic domestication, crop evolution and fertility deprivation. The center of garlic domestication is located in the Irano-Turanian region (Central Asia and Afghanistan). In this region, garlic bulbs were collected from nature and introduced into cultivation ca. 10,000 years ago. Ca. 3000 ago, primitive garlic landraces were introduced to the Mediterranean region, India and China. Crop evolution in various climates resulted in stress adaptations, changes in plant architecture and loss of seed reproduction.

**Figure 10 ijms-24-16777-f010:**
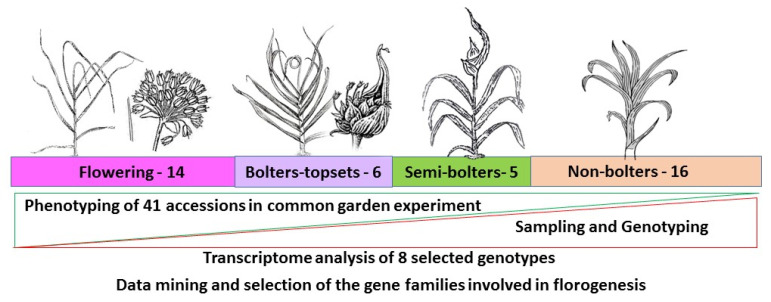
Schematic representation of the experimental design. The phenotypic groups differ in their plant architecture, leaf arrangement, and inflorescence appearance. Fourteen bolting and flowering genotypes produced an inflorescence containing developed flowers and seeds; in six bolting genotypes, the inflorescence contained small topsets, but the flowers were aborted; the five semi-bolting genotypes produced short flower stems and a few large topsets, but no flower meristems and flowers; and the sixteen non-bolting genotypes produced only green leaves.

## Data Availability

The sequencing data were deposited in the NCBI Sequence Read Archive (SRA) database as bioproject PRJNA1004898 [SAMN36965703; SAMN36965704; SAMN36965705; SAMN36965706; SAMN36965707; SAMN36965708; SAMN36965709; SAMN36965710].

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
