# Peer review of "Deprivation of Sexual Reproduction during Garlic Domestication and Crop Evolution"

_ijms, 2023, doi:10.3390/ijms242316777_

Round 1
Reviewer 1 Report
Comments and Suggestions for Authors
In manuscript “Deprivation of sexual reproduction during garlic domestication and crop evolution”, Einat Shemesh-Mayer et al. have undertaken a commendable effort in investigating mechanisms behind garlic crop evolution and have presented a well-structured study. The authors have performed a comprehensive genomics and transcriptome analysis of 41 garlic accessions and identified the potential pathways, genes and SNPs result in the difference. However, I would like to suggest some revisions to enhance the clarity of manuscript. Overall, the work shows promise, and with the suggested revisions, I believe it has the potential to make a valuable contribution to the field.
Major issues:
1. Is the phylogenetic tree in Figure 1. a generated based on variant or transcriptome profiles? Or both? What are the numbers on top of the plot in Figure 1.b?
2. In section 2.3, why cluster B and C are excluded from further analysis? As the authors stated that reference genome is most similar to cluster E, why cluster E not used as the basic one?
3. Proper x and y labels should be added in Figure 4.a. And a, b labels are missing from the figure. Same problem for Figure 5 and 6.
4. Are all pathways analyzed presented in Figure 4, 5 and 6? or only the significant ones, please clarify in the legend or context. Also, what does the color of dots indicate?
5. Numbers on Figure 7 b and d heatmap are not readable, especially for the red cells. Please provide higher quality figures.
6. Could the authors provide one supplementary table for the enriched pathways from all comparisons? If already provided, please ignore this comment. I can only access supp table 1.
7. Will the data be deposited to public database? If so, please provide the accession number in the manuscript.
Minor issues:
1. In line 191, “GO algorithms”, GO is not an algorithm, it’s collection of pathways.
2. In figure 7 b and d, are the heatmaps showing the FPKM values or 1-pearson? And it’s correlation between what?
3. In line 382, “(1) Bolting, inflorescence contains developed flowers;” is it flowering? As shown in the figure.
4. Title of section 3.2 should be genome sequencing, instead of genome analysis.
5. How are the low and high quality determined in section3.3? Authors could mention the threshold in the method.
6. Minor typo issue in line 466, “10.000 years ago” should be “10,000 years ago”.
Author Response
Thank you very much for your help in manuscript improving. Our detailed response to your comments:
Major issues:
- Is the phylogenetic tree in Figure 1. a generated based on variant or transcriptome profiles? Or both? What are the numbers on top of the plot in Figure 1.b?
Figure 1. is generated based on variants in genome. Included in the legend. In 1b - the population structure – upper line represents ancestry fraction estimation. Added to the Fig.
- In section 2.3, why cluster B and C are excluded from further analysis? As the authors stated that reference genome is most similar to cluster E, why cluster E not used as the basic one?
Clusters A, B and C belong to the same morpho-biological type and represent the flowering genotypes of garlic. Therefore, we selected Cluster A as a basic group that also represents both clusters B and C. Added to the text
- Proper x and y labels should be added in Figure 4.a. And a, b labels are missing from the figure. Same problem for Figure 5 and 6.
The REVIGO algorithm provides visualization of the biological process. The following text was added to the legends of the Figs. 4,5 and 6: The multidimensional scaling (MDS) technique is utilized to reduce the dimensionality of semantic similarities among GO terms related to biological processes. The axes in the plot have no intrinsic meaning. Semantically similar GO terms remain close together in the plot. Bubble size represents the p-values of the GO term; bubble color represents the number of annotations for GO term.
- Are all pathways analyzed presented in Figure 4, 5 and 6? or only the significant ones, please clarify in the legend or context. Also, what does the color of dots indicate?
The most significant are represented in the image, others are provided in the Supplementary Table 3. Bubble size represents the p-values of the GO term; bubble color represents the number of annotations for GO term.
- Numbers on Figure 7 b and d heatmap are not readable, especially for the red cells. Please provide higher quality figures.
Figs 7 and 8 were completely re-done for the better presentation.
- Could the authors provide one supplementary table for the enriched pathways from all comparisons? If already provided, please ignore this comment. I can only access sup table 1.
Yes, we added a supplementary Table S3
- Will the data be deposited to public database? If so, please provide the accession number in the manuscript.
Yes, it was deposit. Statement added at the end of the manuscript.
Minor issues:
- In line 191, “GO algorithms”, GO is not an algorithm, it’s collection of pathways.
Changed to "enrichment analysis."
- In figure 7 b and d, are the heatmaps showing the FPKM values or 1-pearson? And it’s correlation between what?
The expression values were computed as FPKM based on transcriptome analyses in each phenotypic group. FPKM values were used to create a heat map. The color scale on the heat map ranges from blue (low value) to red (high value).
- In line 382, “(1) Bolting, inflorescence contains developed flowers;” is it flowering? As shown in the figure.
Corrected in the text
- Title of section 3.2 should be genome sequencing, instead of genome analysis.
Corrected, thank you
- How are the low and high quality determined in section3.3? Authors could mention the threshold in the method.
The technical details are quoted and therefore we did not provide them in the text.
Briefly, SAMtools incorporating probabilistic approach , first samtools mpileup collects in-formation stored in input BAM to computes the likelihood of data given for each possible genotype. Next, bcftools applies the prior probability and performs the actual SNP calling. The output file is a VCF file contains only the variants in which the subject has at least one allele different from the reference allele.
Before any downstream analysis of our VCF files we filtered out variants with low confidence, to maximize retained signal while minimizing systematic error:
GQ parameter is the Genotype Quality of each individual - represents the Phredscaled confidence that the genotype assignment is correct, derived from the genotype probability. GQ of 3 means there's a 50 percent chance that the call is incorrect.
Another parameter that could be checked for is the depth of the reading at each locus, indicated as DP in a VCF file. We set DP > 3 in each individual.
We want the probability that the call is not a variant to be below 0.01 (99% probability that it is a variant), then we ask that ???? ≥ 20
- Minor typo issue in line 466, “10.000 years ago” should be “10,000 years ago”.
Corrected, thank you

Reviewer 2 Report
Comments and Suggestions for Authors
Dear Authors,
The submitted manuscript titled „Deprivation of sexual reproduction during garlic domestication and crop evolution” contains very interesting results, which might interest the international audience. Nevertheless I have found some flawns, which (in my opinion) should be corrected before an eventual publication. Please, find them below:
1. Introduction
• In my opinion in chapter Introduction the justification of choice Allium sativum L for investigations should be added.
• Moreover, I suggest the enlargment of characteristics of garlic. It should contain data about morphology, way of reproduction, lifespan, habitat affiliation and range of localities.
• Also, the information about current stateof knowledge on deprivation of sexual reproduction in garlic with brief review of literature would be stronly needed.
• The chapter Introduction should be endede by specific goals of investigations or working hypotheses.
2. I suggest to compare the obtained results with other literature sources reffering to other species such as e.g. Allium vineale.
3. Figures 4-7 are illegibile.
Author Response
Thank you very much for your suggestions. We related to all of them, and I believe that the quality was improved. Please find our responses in bold:
- Introduction
- In my opinion in chapter Introduction the justification of choice Allium sativum L for investigations should be added.
- Moreover, I suggest the enlargment of characteristics of garlic. It should contain data about morphology, way of reproduction, lifespan, habitat affiliation and range of localities.
- Also, the information about current stateof knowledge on deprivation of sexual reproduction in garlic with brief review of literature would be stronly needed.
The following information added:
In the framework of the garlic breeding program in Israel, we were able to collected and reproduced hundreds of genotypes from various climatic zones varying in their re-productive traits. Large number of flowering genotypes producing viable reproductive or-gans and seeds was found in Central Asia Only a few garlic genotypes are able to pro-duce viable reproductive organs and seeds [11-13]. Further morpho-physiological studies showed that iIn some garlic plants flower initiation and bolting (elongation of flower scape) are completely impaired, others develop small topsets (bulblets) within the inflorescence which compete with flowers for nutrients. Moreover, even if individual flowers succeed in differentiating, seed production might be compromised by male or female sterility, tapetum degeneration, and pollen abortion [11-13]. Only a few garlic genotypes are able to produce viable reproductive organs and seeds [11-13].
Similar to many other perennial monocots, Allium species possess a large genome size (7–32 Gb) (Ricroch et al. 2005). Despite its domestication, garlic has maintained its ploidy level (2n = 2x = 16), and the diploid garlic nuclear genome is estimated at 15.9 Gbp, 32 times larger than the genome of rice (Fritsch and Friesen 2002). Garlic is a diploid species with eight chromosomes. Since Allium genomes are very large and their Therefore, full genome sequence is challenging, and previous garlic genetic studies were based on transcriptome catalogs and candidate gene analysis. Generation of transcriptome catalogs (Sun et al. 2012; Kamenetsky et al. 2015) and resulted in the development of simple sequence repeats (SSRs) that can be used for genetic studies, mapping, and fingerprinting (Liu et al.2015). Organ‐specific transcriptome analysis displayed significant variation in gene expression between different plant organs, while the highest number of specific reads was found in the inflorescences and flowers
- The chapter Introduction should be endede by specific goals of investigations or working hypotheses.
Added to the Introduction:
We postulate that loss of flowering ability during garlic crop evolution is associated not only with specific alterations in the flowering genes, but with large and multiple mutations in the entire genome landscape.
- I suggest to compare the obtained results with other literature sources reffering to other species such as e.g. Allium vineale.
Unfortunately, we did not find any information on genomic or transcriptome analyses of A. vineale
- Figures 4-7 are illegibile.
The Figure quality was improved, Figs 7 and 8 redone.

Reviewer 3 Report
Comments and Suggestions for Authors
In my opinion, the research carried out on 41 garlic genotypes at the phenotypic, genomic and transcriptomic level is interesting. The research seems to be well done, although the manuscript needs to be improved before publication. Please see the comments in the attached document. In summary, at the level of materials and methods, all details should be included, including agronomic, genetic, statistical aspects, replicates, etc. At the results level, the legends and text of some figures should be improved and enlarged, and it should be ensured that all methodological details are explained in the Materials and methods section. The Introduction should include more background if possible, and a separate discussion would be desirable.

In my opinion, some minor editing of the English language is required.
Author Response
Thank you very much for your detailed analysis and most helpful suggestions. I revised your comments in the .pdf file, and our corrections are included. you can also see them in the manuscript version with Track Changes. Our short main responses are provided below, in bold:
In my opinion, the research carried out on 41 garlic genotypes at the phenotypic, genomic and transcriptomic level is interesting. The research seems to be well done, although the manuscript needs to be improved before publication. Please see the comments in the attached document.
Thank you very much for these comments! I feel that they really helped to improve the manuscript. For the detailed changes please refer to the version with Trach Changes.
In summary, at the level of materials and methods, all details should be included, including agronomic, genetic, statistical aspects, replicates, etc.
The details included in all sections of the Materials and Methods
At the results level, the legends and text of some figures should be improved and enlarged, and it should be ensured that all methodological details are explained in the Materials and methods section.
Fig. 7 and 8 were redone completely. In other Figs, quality was improved. Technical clarifications were added to the Fig legends. I double-checked that all details are included in Materials and methods.
The Introduction should include more background if possible, and a separate discussion would be desirable.
Introduction was enriched with more information. We still prefer to combine Results and Discussion in order to avoid redundant wording and repetitions.
